# Adaptive Compensatory Neurophysiological Biomarkers of Motor Recovery Post-Stroke: Electroencephalography and Transcranial Magnetic Stimulation Insights from the DEFINE Cohort Study

**DOI:** 10.3390/brainsci14121257

**Published:** 2024-12-15

**Authors:** Guilherme J. M. Lacerda, Fernanda M. Q. Silva, Kevin Pacheco-Barrios, Linamara Rizzo Battistella, Felipe Fregni

**Affiliations:** 1Neuromodulation Center and Center for Clinical Research Learning, Spaulding Rehabilitation Hospital and Massachusetts General Hospital, Harvard Medical School, Boston, MA 02138, USA; guilherme.lacerda@hc.fm.usp.br (G.J.M.L.);; 2Instituto de Medicina Física e Reabilitação, Hospital das Clínicas, Faculdade de Medicina, Universidade de São Paulo, São Paulo 04101-300, SP, Brazil; 3Vicerrectorado de Investigación, Unidad de Investigación para la Generación y Síntesis de Evidencias en Salud, Universidad San Ignacio de Loyola, Lima 15023, Peru; 4Departamento de Medicina Legal, Bioética, Medicina do Trabalho e Medicina Física e Reabilitação, Faculdade de Medicina, Universidade de São Paulo (FMUSP), São Paulo 01246-903, SP, Brazil

**Keywords:** stroke, rehabilitation, motor recovery, neurophysiological biomarkers, electroencephalography (EEG), resting-state EEG, theta/alpha ratio (TAR), transcranial magnetic stimulation (TMS)

## Abstract

Objective: This study aimed to explore longitudinal relationships between neurophysiological biomarkers and upper limb motor function recovery in stroke patients, focusing on electroencephalography (EEG) and transcranial magnetic stimulation (TMS) metrics. Methods: This longitudinal cohort study analyzed neurophysiological, clinical, and demographic data from 102 stroke patients enrolled in the DEFINE cohort. We investigated the associations between baseline and post-intervention changes in the EEG theta/alpha ratio (TAR) and TMS metrics with upper limb motor functionality, assessed using the outcomes of five tests: the Fugl-Meyer Assessment (FMA), Handgrip Strength Test (HST), Pinch Strength Test (PST), Finger Tapping Test (FTT), and Nine-Hole Peg Test (9HPT). Results: Our multivariate models identified that a higher baseline TAR in the lesioned hemisphere was consistently associated with poorer motor outcomes across all five assessments. Conversely, a higher improvement in the TAR was positively associated with improvements in FMA and 9HPT. Additionally, an increased TMS motor-evoked potential (MEP) amplitude in the non-lesioned hemisphere correlated with greater FMA-diff, while a lower TMS Short Intracortical Inhibition (SICI) in the non-lesioned hemisphere was linked to better PST improvements. These findings suggest the potential of the TAR and TMS metrics as biomarkers for predicting motor recovery in stroke patients. Conclusion: Our findings highlight the significance of the TAR in the lesioned hemisphere as a predictor of motor function recovery post-stroke and also a potential signature for compensatory oscillations. The observed relationships between the TAR and motor improvements, as well as the associations with TMS metrics, underscore the potential of these neurophysiological measures in guiding personalized rehabilitation strategies for stroke patients.

## 1. Introduction

Stroke is the second leading cause of death worldwide and the primary cause of disability, impacting various functional domains such as mobility, touch, pain, cognition, and mood [1]. The incidence of stroke is increasing due to an aging global population, with approximately 16.9 million new cases each year [2]. Despite scientific advances improving rehabilitation outcomes over the past decade, many stroke survivors still experience reduced mobility and severe chronic disability, particularly in the upper limbs, after a year of inpatient rehabilitation [3]. This highlights the ongoing challenge of implementing effective rehabilitation strategies for stroke survivors, which remains in its incipient stages.

In the context of stroke recovery, which has significant heterogenicity among patients, further research is essential to understand the disability mechanisms and enhance patient-focused rehabilitation, developing predictors of outcomes for various interventions [4]. Since rehabilitation marks a complex neurobiological phase, identifying reliable biomarkers is vital [5]. Many biomarkers of stroke recovery have been discussed in the literature, such as those derived from Electroencephalography (EEG) and Transcranial Magnetic Stimulation (TMS), either in isolation or combined. For instance, recent studies have demonstrated that EEG shows potential in predicting upper limb sensorimotor recovery [6,7,8,9,10] and TMS shows potential for upper extremity motor function improvement, both in post-stroke patients [9,11,12]. Therefore, the EEG and TMS-metrics could serve as an effective brain biomarker for upper limb motor function recovery.

Moreover, the role of the EEG and TMS metrics in understanding and predicting stroke recovery underscores their importance in stroke rehabilitation. A 2017 systematic review identified the motor-evoked potential (MEP) derived from TMS as a brain biomarker for improved upper limb motor outcomes [7]. A 2023 systematic review further highlighted EEG’s potential in predicting upper extremity sensorimotor recovery, emphasizing its ability to capture neural alterations such as interhemispheric imbalance and disruptions in beta oscillatory activity, which are closely linked to motor recovery outcomes [13]. However, despite these advances, significant gaps remain in the identification of reliable biomarkers due to methodological limitations and the complexity of interpreting interactions between biomarkers and clinical measures [5]. For instance, recent research employing multivariate models revealed a specific neural signature in which a higher EEG theta/alpha ratio (TAR) in the lesioned hemisphere combined with increased MEP amplitude in the non-lesioned hemisphere correlates with poorer upper limb motor function [14]. This finding suggests a potential neural signature of brain compensation where lower frequencies of EEG power are heightened in the lesioned hemisphere, and corticospinal excitability is reduced in the non-lesioned hemisphere. Consequently, there is a clear need for more comprehensive studies utilizing the EEG and TMS metrics to enhance our understanding of upper extremity recovery patterns post-stroke.

To advance the knowledge of upper limb recovery post-stroke, our longitudinal study employs a prospective cohort analysis exploring EEG theta/alpha oscillations as potential biomarkers for predicting upper extremity impairment and motor outcomes. We aim to determine how this measure correlates with five motor assessments: the Fugl-Meyer Assessment (FMA), Handgrip Strength Test (HST), Finger Tapping Test (FTT), Nine-Hole Peg Test (9HPT), and Pinch Strength Test (PST) [15]. Our models are designed to capture recovery patterns and biomarker changes, emphasizing the potential of the EEG-TAR in predicting motor impairment and recovery in the upper limbs of stroke patients. We hypothesize that a decreased EEG-TAR will be related to a better functional improvement and this approach could significantly enhance current rehabilitation practices by facilitating the development of personalized treatment strategies.

## 2. Materials and Methods

### 2.1. Participants, Study Design, and Sample Size

This study involves a longitudinal analysis employing baseline and post-interventional data from the “Deficit of Inhibition as a Marker of Neuroplasticity (DEFINE Study) in Rehabilitation: A Longitudinal Cohort Study Protocol” cohort study [16]. This cohort recruited patients from the conventional stroke rehabilitation program at the Instituto de Medicina Física e Reabilitação (IMREA). In the stroke arm of this cohort, 102 participants signed the informed consent, which had been previously approved by the Hospital das Clínicas da Faculdade de Medicina da Universidade de São Paulo Ethics Committee for Research Protocol Analysis (CAAE: 86832518.7.0000.0068). Here, we analyze data from those subjects who underwent a series of clinical and neurophysiological evaluations at two key stages: before starting and after completing the IMREA rehabilitation program.

### 2.2. Inclusion Criteria

From 2020 to 2022, subjects of both sexes, aged 18 years and older, were included if they had a clinical and radiological diagnosis of stroke confirmed by imaging and were clinically stable as verified by a medical evaluation. They also needed to meet the eligibility criteria for the IMREA rehabilitation program. Exclusion criteria included any clinical or social conditions that could interfere with participation in the rehabilitation treatment. Pregnancy was also a criterion for exclusion. All participants provided signed informed consent forms [16].

### 2.3. Exclusion Criteria

Participants were excluded from the study if they had any clinical or social conditions that could interfere with their participation in the rehabilitation treatment. Additionally, pregnancy was considered an exclusion criterion.

### 2.4. Functional Assessments

A trained physician first evaluated a participant’s eligibility, reviewed their medical history, conducted physical examinations, and collected demographic surveys. The data collected were then used to identify covariates for the regression models. Various scales were used to describe the sample and controls for confounding variables in the multivariate statistical model. To evaluate upper limb function, we selected five specific assessments, each targeting a unique aspect of motor performance: The Fugl-Meyer Assessment (FMA) measures motor recovery and coordination in the upper limb, providing a comprehensive evaluation of motor impairment. The Handgrip Strength Test (HST) assesses overall grip strength, reflecting functional hand performance and general muscle strength. The Pinch Strength Test (PST) evaluates fine motor strength required for precision tasks, such as pinching and gripping small objects. The Finger Tapping Test (FTT) measures motor speed and coordination, focusing on fine motor control and rhythm. The Nine-Hole Peg Test (9HPT) assesses dexterity and hand-eye coordination, crucial for everyday manual tasks. These assessments were selected to provide a well-rounded evaluation of upper limb function [15,17]. A more detailed description of each assessment is available in the Appendix A.

### 2.5. Intervention

The rehabilitation interventions provided at IMREA are highly personalized, tailored to the unique needs of each patient. These interventions encompass a multidisciplinary approach, including physical therapy to enhance motor function recovery, strength, and mobility; occupational therapy to improve activities of daily living and fine motor skills; and psychological support to address emotional well-being and coping strategies post-stroke. Additional therapies, such as speech therapy or social support services, may also be incorporated based on the patient’s specific condition and recovery goals. This personalized and comprehensive framework ensures that each patient receives targeted care to optimize their rehabilitation.

### 2.6. Neurophysiological Assessment Methods

#### 2.6.1. Electroencephalography

A skilled clinical neurophysiologist conducted a visual analysis of the EEG data to identify artifacts and any potential clinical abnormalities. This information was subsequently exported and analyzed offline using MATLAB (R2014b, The MathWorks Inc., Natick, MA, USA) and EEGLab [18]. The EEG should will last approximately 45 min, consisting of 25 min for participant and software setup, followed by 10 min of EEG recording. This recording will be split into a resting-EEG condition, with 5 min of open-eye and 5 min of closed-eye observation, and an 8 min task-related EEG condition. The task-related EEG part includes activities such as movement observation, imagery, and execution. The analysis focused on standard frequency bands: delta (2–4 Hz), theta (4–8 Hz), low-alpha (8–10.5 Hz), high-alpha (10.5–13 Hz), alpha (8–13 Hz), low-beta 1 (13–20 Hz), high-beta 2 (20–30 Hz), and beta (13–30 Hz). These frequency bands were measured bilaterally across the frontal, central, and parietal scalp regions and used to calculate the theta/alpha ratio (TAR), which is the simple division of theta band power (4–8 Hz) by alpha band power (8–13 Hz).

#### 2.6.2. Transcranial Magnetic Stimulation

A Magstim Rapid^®^ stimulator (The Magstim Company Limited, Whitland, UK) with a 70 mm figure-of-eight coil was positioned tangentially on the skull at a 45-degree angle to the sagittal line for the TMS assessment [19]. All assessments were performed bilaterally by stimulating the motor cortex, with muscular responses recorded using surface electromyography electrodes placed on the target muscle, the first dorsal interosseous of the hand. To locate the cortical area corresponding to the selected muscle, the vertex (intersection of the nasion–inion lines and zygomatic arches) was identified. Marks were then drawn 5 cm (about 1.97 in) from the vertex towards the ear tragus on both sides in (rMT) the coronal plane, identifying the “hot spot”—the area with the lowest resting motor threshold and the highest motor-evoked potential (MEP) amplitude in the target muscle.

The rMT is defined as the minimum intensity at which a single TMS pulse at the hot spot elicits an MEP with at least a 50 V peak-to-peak amplitude in 50% of attempts. The assessment also included recording ten MEPs at 130% of rMT with a 7 s interval between stimuli (the average of those measures is the MEP130); measuring the silent period (SP), which reflects the temporary suppression of electromyographic activity during a sustained MEP voluntary contraction; assessing short-interval intracortical inhibition (ICI) with two stimuli 2 milliseconds apart, with the conditioned stimulus at 80% rMT and the test stimulus at 130% rMT; and measuring intracortical facilitation (ICF) using the same stimulus values as ICI but with a 10-millisecond interval between the stimuli. Moreover, the ICI is calculated as the MEP after paired-pulse 2 ms/the MEP at baseline, and the SICF is determined by the MEP after paired-pulse 10 ms/the MEP at baseline [20].

### 2.7. Statistical Analysis

In this cohort study, we collected two measurements for each variable: one before and one after the rehabilitation program. We then calculated the differences by subtracting the pre-intervention values from the post-intervention values for each variable. These differences represent our outcomes and are designated as follows: FMA-diff for the Fugl-Meyer Assessment, HST-diff for the Handgrip Strength Test, PST-diff for the Pinch Strength Test, FTT-diff for the Finger Tapping Test, and 9HPT-diff for the Nine-Hole Peg Test. All other variables follow the same naming convention: variables representing the difference between post- and pre-intervention measurements have the suffix “-diff” added to their names. In contrast, variables without this suffix represent baseline (pre-intervention) data.

For the analysis, only complete cases were included. Outcomes were dichotomized into binary variables as they did not satisfy the assumptions necessary for linear regression.

A value of 0 was assigned when the difference between post- and pre-intervention measurements was zero or negative (difference ≤ 0), indicating a deterioration in the patient’s condition. Conversely, a value of 1 was assigned when the difference was positive (difference > 0), indicating an improvement in the patient’s condition. An exception was made for the 9HPT-diff, where a lower time reflects a better performance; hence, the categorization was reversed. In this case, a value of 1 was assigned when the difference was negative (difference < 0), indicating improvement, while a value of 0 was assigned when the difference was zero or positive (difference ≥ 0), indicating a decline in the patient’s condition.

We then investigated two types of associations: first, between the outcomes (FMA-diff, HST-diff, PST-diff, FTT-diff, and 9HPT-diff) and baseline variables measured prior to the intervention; and second, between the same outcomes and the differences in neurophysiological variables, calculated as the post-intervention values minus the pre-intervention values, in the same manner as the outcomes were determined. In both cases, we first performed univariate regression analyses. Subsequently, we used the variables that had a *p*-value < 0.25 in the univariate associations. Finally, we constructed the final multivariate model using a combination of Forward Selection and Backward Elimination, guided by clinical reasoning in the selection of variables. During this stage, a significance level of *p*-value < 0.05 was adopted. It is noteworthy that we thoroughly examined potential confounders such as age, sex, educational level, and others, adjusting the model as necessary. We considered a variable to be a confounder if it changed the beta-coefficient (Log [Odds Ratio]) of another variable by more than 10%. Additionally, given the exploratory characteristic of this study, we investigated region-specific relationships between EEG metrics and motor outcomes, providing preliminary evidence for the distinct roles of the central and parietal regions. The central region, involving the primary motor cortex, was linked to gross motor functions such as strength and speed, assessed by FMA, HST, and FTT. In contrast, the parietal cortex, key for sensorimotor integration and dexterity, was associated with fine motor skills, as reflected in the 9HPT, highlighting its role in precision and coordination.

Statistical significance was determined at *p* < 0.05, and all analyses were conducted using a standard software package, R-Studio Version 2023.06.0+421 (2023.06.0+421).

## 3. Results

### 3.1. Sample Characteristics

Baseline data for 102 post-stroke patients who participated in the cohort study were collected, revealing that the cohort had a mean age of 56 years old, 58 were male (57%), the mean level of education was 11.46 years, and most participants were white, 52 (51%). Most had an ischemic stroke (81%), which was a moderate stroke (mean of National Institute of Health Stroke Scale [NIHSC] of 5) and required a mild level of assistance (Functional Independence Measure [FIM] of 96). Additionally, all five upper limb motor function metrics demonstrated a non-normal distribution. A detailed description of the sample demographic and clinical characteristics at baseline is provided in Table 1.

### 3.2. Dependent Variables Characteristics

Table 2 summarizes the five upper limb motor outcomes, specifically the differences between post- and pre-intervention for each assessment. For FMA-diff, 52.7% of participants had a difference ≤0, while 47.3% had a difference >0. For HST-diff, 57.9% of participants had a difference ≤0, and 42.1% had a difference 0. For PST-diff, 66.7% of participants had a difference ≤0, and 33.3% had a difference >0. For FFT-diff, 61.5% of participants had a difference ≤0, while 38.5% had a difference >0. Lastly, for 9HPT-diff, 72.7% of participants had a difference ≤0, and 27.3% had a difference >0.

### 3.3. Multivariate Analysis

When necessary, we adjusted each model for covariates such as age, sex, time of injury, education level, marital status, stroke type, BMI, smoking history, NIHSS, and SICI. This adjustment allowed us to accurately reveal the associations between the outcomes and the TAR.

### 3.4. Association Between Outcomes and Baseline Neurophysiological Variables

These models were constructed to identify associations between the five outcomes and the baseline variables, adjusted by the patients’ demographic and clinical characteristics (Table 3). For FMA-diff, we found a negative association with the TAR measured in the parietal lesioned hemisphere (OR = 0.22, 95% CI: 0.04–0.82; *p*-Value = 0.04), and a positive association with MEP130 measured in the non-lesioned hemisphere, OR = 1.57 (95% CI: 1.05–2.48; *p*-Value = 0.03). For HST-diff, we found a significant negative association with the TAR measured in the central lesioned hemisphere (OR = 0.20, 95% CI: 0.04–0.75; *p*-Value = 0.0351). For PST-diff, we observed a significant negative association with the TAR measured in the central lesioned hemisphere (OR = 0.19, 95% CI: 0.03–0.78; *p*-Value = 0.0456) and a significant positive association with ICI measured in the non-lesioned hemisphere (OR = 14.62, 95% CI: 1.75–209.10; *p*-Value = 0.025). For FTT-diff, we identified a significant negative association with the TAR measured in the central lesioned hemisphere (OR = 0.08, 95% CI: 0.00–0.56; *p*-Value = 0.0393). For 9HPT-diff, we found a significant negative association with the TAR measured in the central lesioned hemisphere (OR = 0.13, 95% CI: 0.01–0.66; *p*-Value = 0.0371).

### 3.5. Association Between Outcomes and Neurophysiological Variables Differences (Post- and Pre-Intervention)

These models were constructed to identify associations between the outcomes and the differences in the variables (post-intervention minus pre-intervention), adjusted by the patients’ demographic and clinical characteristics (Table 4). For FMA-diff, we found a positive association with the TAR-diff measured in the central lesioned hemisphere (OR = 1.15, 95% CI: 1.02–1.36, *p*-Value = 0.0454). For 9HPT-diff, we found a significant positive association with the TAR-diff measured in the parietal lesioned hemisphere (OR = 1.81, 95% CI: 1.15–3.62; *p*-Value = 0.029). No significant associations were found for HST-diff, PST-diff, or FT-diff.

## 4. Discussion

### 4.1. Main Findings

In our current research, we explored five outcomes: FMA-diff, HST-diff, PST-diff, FT-diff, and 9HPT-diff. We identified a negative association between the outcomes and baseline TAR measured in the lesioned hemisphere, as well as a positive association between the outcomes and the TAR-diffs (post- minus pre-intervention) also measured in the lesioned hemisphere. Additionally, we observed a positive association between baseline MEP130 measured in the non-lesioned hemisphere and FMA-diff. Similarly, a positive association was found between baseline SICI, also measured in the non-lesioned hemi-sphere, and PST-diff.

### 4.2. Association Between Outcomes and Baseline EEG Theta/Alpha Ratio in the Lesioned Hemisphere

The models aimed to identify associations between five motor outcome measures (FMA-diff, HST-diff, PST-diff, FTT-diff, and 9HPT-diff) and baseline variables, while adjusting for patients’ demographic and clinical characteristics. The TAR measured in the lesion hemisphere showed a significant negative association with the five outcomes, indicating that higher baseline TAR values were associated with less improvement across different assessments.

Building upon this, our previous study showed that a higher TAR in the frontal region of the lesioned hemisphere is associated with poorer motor outcomes, as seen in the patients’ lower performance scores. This observation is consistent with prior studies that have identified an increased ratio of [slow oscillations]/[fast oscillations] as a marker of brain injury (DAR: Delta/Alpha and DTABR: (delta + theta)/(alpha + beta)). For instance, Finnigan et al. reported that the DAR and DTABR are reliable indicators for detecting acute ischemic stroke [21]. Furthermore, evidence indicates that low-frequency oscillations in the motor cortex, which relate to motor control tasks, tend to decrease shortly after a stroke and re-emerge during motor recovery [22,23]. Additionally, a transition toward alpha oscillations is associated with improvements in motor abilities [24]. In our previous cross-sectional study, we suggested that a reduced theta/alpha ratio might be associated with improved motor function [14]. In the current study, which includes two measurements (pre- and post-intervention), this hypothesis appears to be supported, as we observed consistent results across all five outcomes. In fact, the TAR has consistently been identified as a significant factor associated with motor outcomes. For example, in our models, we observed that a higher TAR in the lesion hemisphere was negatively correlated with various motor performance measures. Specifically, for FMA-diff, we found that an elevated TAR in the parietal lesion region was associated with poorer motor outcomes. Similarly, the TAR in the central lesion region was negatively associated with HST-diff, PST-diff, FTT-diff, and 9HPT-diff. These consistent findings across multiple outcome measures underscore the importance of the TAR as a potential marker for motor function, suggesting that a higher baseline TAR in the lesion hemisphere is associated with poorer improvements across all five motor outcomes. Finally, our findings solidify the role of TAR as a reliable predictor of motor outcomes in stroke patients, offering valuable insights for both research and clinical practice. Furthermore, the relationships between TAR and the different functional assessments also highlight variations in the aspects of motor function being measured. For instance, the 9HPT and FTT assess motor speed and dexterity, while the HST and PST are more focused on strength-related performance. This distinction suggests that the influence of TAR might manifest differently depending on the motor domain being assessed. Exploring these nuanced relationships further could provide deeper insights into how TAR relates to specific functional abilities and recovery trajectories.

### 4.3. Association Between Outcomes and EEG Theta/Alpha Ratio (Differences Post- Minus Pre-Intervention) in the Lesioned Hemisphere

The models were constructed to identify associations between the motor outcomes and the TAR-diff in the lesioned hemisphere, while adjusting for patients’ demographic and clinical characteristics. For FMA-diff, a positive association was found with TAR-diff, indicating that an increase in this ratio was associated with improved motor function, as reflected by higher FMA-diff scores. Similarly, a significant positive association was observed between 9HPT-diff and TAR-diff, suggesting that increases in the TAR in the parietal lesion region were linked to a better motor performance. These findings imply that changes in the TAR post-intervention may serve as indicators of motor recovery, particularly in the central and parietal regions of the brain. Based on the previous topic, studies have identified an increased ratio of [slow oscillations]/[fast oscillations] as a marker of brain injury. Additionally, evidence indicates that low-frequency oscillations in the motor cortex, which are related to motor control tasks, tend to decrease shortly after a stroke and re-emerge during motor recovery [22,23]. Moreover, a transition toward alpha oscillations is associated with improvements in motor abilities. This aligns with our results, where we found that higher differences in the TAR were associated with greater improvements in both FMA-diff and 9HPT-diff, suggesting that these changes in the TAR could reflect a more robust recovery process in motor function. This finding suggests that the transition from delta oscillations to alpha oscillations, as represented by an increase in the theta/alpha ratio (TAR), may serve as a potential biomarker for predicting improvements in motor outcomes. The observed increase in the TAR could reflect underlying neurophysiological processes associated with recovery, such as the reorganization of neural networks or the restoration of motor control pathways. As such, monitoring changes in the TAR during rehabilitation could provide valuable insights into the effectiveness of interventions and the trajectory of motor function recovery.

It is worth mentioning that we did not find statistically significant associations between TAR-diff in the lesioned hemisphere and HST-diff, PST-diff, and FT-diff. This lack of association may be due to these three outcomes being more closely related to gross motor movements, which might not be as sensitive to changes in the theta/alpha ratio as fine motor skills. Furthermore, the differential associations observed between TAR-diff and motor outcomes highlight the varying sensitivity of these assessments to neurophysiological changes. For example, FMA-diff and 9HPT-diff primarily assess fine motor control and coordination, which may be more directly influenced by changes in the TAR. In contrast, outcomes like HST-diff and PST-diff, which focus on strength-related measures, may rely less on the neural oscillatory dynamics captured by the TAR. These differences underscore the importance of tailoring the selection of functional assessments to the specific neurophysiological mechanisms under investigation and suggest that the TAR may serve as a more robust biomarker for fine motor recovery compared to gross motor outcomes.

### 4.4. Association Between Outcomes and Baseline TMS Metrics in the Non-Lesioned Hemisphere

Additionally, we observed a positive association between baseline MEP130 measured in the non-lesioned hemisphere and FMA-diff. This suggests that higher baseline MEP values are associated with greater improvements in motor function. Similarly, a positive association was found between baseline SICI, also measured in the non-lesioned hemisphere, and PST-diff, indicating that a lower inhibition (as reflected by higher SICI amplitude values) may be linked to better motor recovery outcomes.

Our previous study demonstrated that increased MEP amplitude in the non-lesioned hemisphere correlates with improved motor function [14]. These relationships were consistent across all five motor assessments, reinforcing the idea that neurophysiological measures, such as MEP amplitude, may serve as reliable biomarkers for recovery. Indeed, in the current study, a higher baseline MEP130 of the motor threshold is associated with a greater score for FMA-diff, suggesting that increased activity in the non-lesioned hemisphere at baseline is linked to higher odds of improving motor skills, as measured by the Fugl-Meyer Assessment. Moreover, the finding of a positive association between SICI and PST-diff is consistent with this idea, as higher SICI values indicate less inhibition. Therefore, we could state that the lower the inhibition in the non-lesioned hemisphere, the higher the odds of improving motor skills, as measured by the Pinch Strength Test.

### 4.5. Strengths and Limitations

One limitation would be that the relatively small sample size may have negatively impacted our results’ statistical power and precision, underlining the need for further larger longitudinal and interventional studies to establish more accurate causal associations. Additionally, it is important to acknowledge that the DEFINE cohort study was conducted in a Latin American center, which may influence the external validity of our findings. Furthermore, the dichotomization of continuous outcomes, while pragmatic for addressing data distribution challenges, may have reduced granularity and masked nuanced relationships, warranting a cautious interpretation of the results. It should be mentioned that another limitation of this study is the lack of stroke location data, which may influence EEG and recovery outcomes.

On the other hand, a key strength of our research lies in its comparison of neuro-physiological measures with the differences in pre- and post-intervention values across five motor assessments within a stroke population. This approach not only enriches our understanding of the clinical validity of EEG and TMS measures, as neurophysiological biomarkers of upper limb motor recovery, but also provides a foundation for evaluating the impact of changes in EEG and TMS metrics as potential surrogate outcomes for further studies. Additionally, the longitudinal nature of our analysis offers a more dynamic representation of post-stroke recovery.

### 4.6. Future Perspectives: EEG Biomarkers and Emerging Parameters for Analysis

In our study, the TAR has been identified as a potential biomarker for motor recovery, reflecting cortical reorganization processes during rehabilitation. These findings align with research on event-related desynchronization (ERD) during action observation, which has been highlighted as an early predictor of motor recovery in subcortical stroke [25]. Specifically, alpha ERD during AO in the affected parietal and central regions has been correlated with better functional outcomes, such as improvements in FMA scores at 12 weeks. Similarly, the TAR captures broader oscillatory dynamics that reflect the brain’s capacity for functional reorganization. While our study focuses on the TAR’s predictive value, integrating ERD into future investigations alongside the TAR may enhance our understanding of recovery processes, particularly in characterizing the engagement of motor-related regions even in patients with severe impairments.

Additionally, as emphasized in [26], combining kinematic measures with EEG biomarkers provides complementary insights into post-stroke recovery. Kinematic parameters, such as movement duration and smoothness, have been strongly correlated with FMA scores, indicating their relevance as measures of motor recovery. Integrating kinematic data into studies focusing on the TAR could deepen our understanding of the neurophysiological and biomechanical correlations with motor recovery. This multimodal approach has the potential to optimize personalized rehabilitation strategies by leveraging insights from both neurophysiological and movement-related parameters.

## 5. Conclusions

Our study demonstrated that the TAR in the lesioned hemisphere consistently correlated with upper motor function recovery post-stroke across various prediction models. Specifically, a lower baseline TAR was associated with better outcomes in motor assessments, including the FMA-diff, HST-diff, PST-diff, FT-diff, and 9HPT-diff. These findings underscore the potential of the TAR as a reliable biomarker for predicting upper limb motor recovery post-stroke. Patients with a lower baseline TAR may possess a greater capacity for motor function improvement during rehabilitation. Additionally, our results revealed that an increase in the TAR-diff was positively associated with greater improvements in motor outcomes. This suggests that as the TAR increases after the rehabilitation process, better motor function outcomes are observed, further supporting the role of TAR as a dynamic marker of recovery. Our findings highlight the potential for the EEG theta/alpha ratio (TAR) and TMS metrics as neurophysiological biomarkers to predict motor recovery in stroke patients. Clinically, these biomarkers can provide valuable insights into individual recovery trajectories, enabling more personalized rehabilitation strategies. For example, the association between TAR and motor outcomes suggests its utility as a marker for identifying patients who may benefit from targeted interventions to enhance compensatory oscillations in the lesioned hemisphere. Moreover, metrics such as MEP amplitude and SICI in the non-lesioned hemisphere could help guide the use of neuromodulatory techniques like repetitive TMS (rTMS) or paired associative stimulation (PAS) to optimize motor recovery. These biomarkers could also aid in stratifying patients for clinical trials, ensuring that interventions are tailored to specific neurophysiological profiles.

## Figures and Tables

**Table 1 brainsci-14-01257-t001:** Demographic and clinical features at baseline for the 102 post-stroke patients included in the study.

Patient Characteristics	Total (N = 102)
Age, mean (SD)	56.02 (13.7)
Sex, N (%)	
Male	58 (58.86)
Female	44 (43.14)
Education years, mean (SD)	11.46 (5.83)
BMI, mean (SD)	26.77 (5.17)
Smoking history, N (%)	
Yes	38 (37.2%)
No	54 (53.0%)
Missing data	10 (9.8%)
Marital status, N (%)	
Married	55 (54.06)
Not married	47 (45.94)
Race, N (%)	
Asian	7 (6.90)
Black	10 (9.70)
More than one race	26 (25.5)
White	52 (51)
Unknown or not reported	7 (6.90)
Stroke type, N (%)	
Ischemic	83 (81.37)
Hemorrhagic	12 (11.76)
Ischemic with hemorrhagic transformation	7 (6.87)
NIHSS, median (IQR)	5 (2–8)
Time since the injury (in months), median (IQR)	12.3 (5.29–19.38)
Patient age at the stroke), median (IQR)	56.15 (44.64–67.36)
Hand dominance aligned with the paretic side, N (%)	
Yes	91 (58.18)
No	42 (41.82)
Impairment of the thenar region on the paretic side, N (%)	
Yes	80.58
No	11 (19.42)
Presence of spasticity, N (%)	
Yes	71 (69.6%)
No	19 (18.6%)
Missing data	12 (11.8%)
CPM on the paretic side, median (IQR)	1.48 (0.65–3.44)
FIM score, median (IQR)	96 (77.75–106.0)

Abbreviations: NIHSS: National Institute of Health Stroke Scale; FIM: Functional Independence Measure; SD: Standard deviation); IQR: Interquartile range; CPM: Conditioned Pain Modulation.

**Table 2 brainsci-14-01257-t002:** Differences between motor function assessments after and before the rehabilitation program.

FMA-Diff	Number of Observations	%
difference ≤ 0	29	52.7
difference > 0	26	47.3
HST-diff		
difference ≤ 0	33	57.9
difference > 0	24	42.1
PST-diff		
difference ≤ 0	38	66.7
difference > 0	19	33.3
FFT-diff		
difference ≤ 0	24	61.5
difference > 0	15	38.5
9HPT-diff		
difference ≤ 0	40	72.7
difference > 0	15	27.3

**Table 3 brainsci-14-01257-t003:** Association between outcomes and baseline neurophysiological variables.

Multivariable Analysis	OR	95% CI	*p*-Value	N
FMA-diff ^a^				48
EEG TAR (parietal lesion side)	0.22	0.04–0.82	0.040	
TMS MEP 130 (non-lesion side)	1.57	1.05–2.48	0.034	
HST-diff ^b^				55
EEG TAR (central lesion side)	0.20	0.04–0.75	0.035	
PST-diff ^b^				53
EEG TAR (central lesion side)	0.19	0.03–0.78	0.046	
TMS SICI (non-lesion side)	14.62	1.75–209.10	0.025	
FT-diff ^b^				39
EEG TAR (central lesion side)	0.08	0.00–0.56	0.04	
9HPT-diff ^b^				53
EEG TAR (central lesion side)	0.13	0.01–0.66	0.037	

^a^ Adjusted by FMA baseline, marital status, education level, and smoking history. ^b^ Adjusted by level of education and BMI. Abbreviations: FMA: Fugl-Meyer Assessment; HST: Handgrip Strength Test; PST: Pinching Strength Test; FT: Finger Tapping; 9HPT: Nine Hole Peg Test; EEG: Electroencephalogram; TMS: Transcranial Magnetic Stimulation; TAR: theta over alpha ratio; MEP: motor evoked-potential; SICI: Short Intracortical Inhibition; OR: Odds Ratio; CI: Confidence Interval; BMI: Body Mass Index.

**Table 4 brainsci-14-01257-t004:** Association between outcomes and neurophysiological variables differences.

Multivariable Analysis	OR	95% CI	*p*-Value	N
FMA-diff ^a^				41
EEG TAR (central lesion) difference	1.15	(1.02–1.36)	0.045	
9HPT-diff ^b^				41
EEG TAR (parietal lesion side) difference	1.81	(1.15–3.62)	0.029	

^a^ Adjusted by age, sex, NIHSS, and stroke type. ^b^ Adjusted by NIHSS, time since injury, EEG TAR frontal lesion side, and EEG TAR central lesion side. Abbreviations: FMA: Fugl-Meyer Assessment; 9HPT: Nine Hole Peg Test; EEG: Electroencephalogram; TMS: Transcranial Magnetic Stimulation; TAR: theta over alpha ratio; OR: Odds Ratio; CI: Confidence Interval; BMI: Body Mass Index; NIHSS: National Institute of Health Stroke Scale.

## Data Availability

The original contributions presented in the study are included in the article and Appendix A, further inquiries can be directed to the corresponding author.

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
