# Peer review of "Adaptive Compensatory Neurophysiological Biomarkers of Motor Recovery Post-Stroke: Electroencephalography and Transcranial Magnetic Stimulation Insights from the DEFINE Cohort Study"

_brainsci, 2024, doi:10.3390/brainsci14121257_

Round 1

Reviewer 1 Report

Comments and Suggestions for Authors

The paper by Guilherme J M Lacerda et al. is a cohort study evaluating the utility of EEG and NIBS biomarkers to predict post-stroke motor recovery. The topic is of great interest, the sample size is adequate, and the results are promising. However, the study has some significant limitations in terms, above all, of the quality of the writing, with many omissions and little care taken in describing the study procedure. Specifically: 

- Please always use the full name first and then the acronym and justify the importance of what you mention (e.g. NIHSS, rMT); 

- Please add the relevant citations in the text where it currently appears (ref)(ref);

- In the Introduction, many aspects are only hinted at, and that would instead be important to elaborate on, and others are not introduced at all and should instead be explained. For example, the Authors later talk about (S)ICI (please call it SICI as is generally done in the current literature), ICF, and SP; what are they? Why should it be important in stroke patients recovery? 

- Also in the Introduction, I suggest not reporting ‘combination’ of TMS and EEG, because TMS-EEG is a specific technique, whereas, in this case, the Authors just combine biomarkers provided by both methods; 

- The Introduction cites some work that is now dated, so I suggest citing more recent reviews on the relationship between EEG measures and upper limb motor recovery in stroke patients. This is also particularly important for the Discussion, in which it would also be appropriate to mention other EEG measures of interest in this context (e.g., cortico-muscular coherence, BSI, DTABR) to better explore the relationships between these EEG-provided metrics versus those described by the Authors; 

- The Methods have, in my opinion, serious shortcomings; it is unclear when exactly the pre-treatment and post-treatment assessments of interest were carried out and, most importantly, what this treatment was; was it the same for all patients? What did it involve? How was it structured? It would be essential to describe it and also to add a timeline of the study; 

- Also, in the Methods, the exclusion criteria are listed twice, and the criteria for exclusion from TMS never appear; furthermore, were patients without MEP in the affected hemisphere included? Were there cut-offs for the severity of motor deficit in the upper limb? And cognitive? If so, assessed how?

- Again, it is not clear whether the patients included were first or stroke patients or whether patients with stroke recurrence were included. Were patients with cerebellar stroke included? Or with bilateral stroke? If so, the implications should be considered; 

- It is not clear to me whether the TMS measures were acquired only from the healthy hemisphere or also from the affected hemisphere. Furthermore, it is unclear why the MEP was acquired at 130% rMT. Was an amplitude close to the millivolt intended? Why? What are the possible implications of this choice?

- In Methods, it would be important to justify the choice of the specific functional assessments chosen. What do they measure respectively?

- In the Results, in my opinion it would be important to carry out stratification of analyses, e.g. ischaemic vs haemorrhagic, cortical vs subcortical, chronic vs subacute phase, etc. Would it be possible to perform linear models that also take into account the various information (e.g. age, NIHSS, comorbidities, time since stroke, etc.) considered?

- Furthermore, also in the Results, the relationships between TMS measures (e.g. rMT, SICI, ICF) between the two hemispheres (if evaluated) with respect to pre- and post-intervention outcomes of interest should be better considered and described in my opinion; 

- In the Discussion, it would be crucial to further explore, both in terms of EEG and TMS, the relationships with the different functional assessments. For example, some of the measures chosen measure precision grip, others are more ‘coarse’ assessments of overall motor function; it would be important to explore these differences further; 

- In the Discussion I suggest to briefly explore, considering the focus on EEG, also neurophysiological measures related to action observation, such as event-related desynchronization during action observation which seems to be an early predictor of recovery in subcortical stroke. This could be crucial to further investigate the neurophysiological correlates of the ability to predict rehabilitation outcome; 

- Also in the Discussion, another aspect that might be interesting to evaluate is that of kinematics, again with a view to providing a comparison between different but complementary methods to characterise these patients. For example, multimodal studies to explore upper limb motor recovery after stroke have combined EEG and Kinematics measures and identified interesting metrics that could be cited in the Discussion to further investigate the functional correlates of the relation between EEG measures and upper limb motor recovery in stroke patients (e.g. doi:10.1177/15500594231209397).

- Finally, the Discussion should, if possible, suggest which categories of patients are those in which the metrics of interest are most useful in predicting recovery, e.g. whether in ischaemic or haemorrhagic or subacute vs. chronic. Furthermore, based on the rehabilitation treatment performed (e.g. traditional? robotic? VR?) it would be useful to assess which type of treatment can predict response. 

Reviewer 2 Report

Comments and Suggestions for Authors

The study aimed to investigate the role of Electroencephalography (EEG) and Transcranial Magnetic stimulation (TMS) on stroke patients' recovery. According to the study, higher slow waves (EEG Theta/Alpha Ratio) on the lesion side were associated with poor recovery of motor function.

Introduction

1. The study hypothesis is decreased EEG-TAR will lead to functional improvement. Previous studies have confirmed that higher TAR in the lesion hemisphere is correlated with poor upper limb motor function. The hypothesis was confirmed. Lesion side have more slow waves and higher TAR in EEG is a natural condition, which cannot be a hypothesis.

Methods

2. It is critical to describe the study period.

3. The interval between stroke onset and EEG study is critical. When performing EEG study needs to be described.

4. 2.2 Inclusion Criteria “Exclusion criteria include any clinical or social conditions that could interfere with participation in the rehabilitation treatment. Pregnancy was also a exclusion criteria."  

This sentence is the same as 2.3 Exclusion criteria. It is not necessary.

5. The inclusion criteria should include how long after stroke onset. EEG may change in different stages after stroke.

6. How long after rehabilitation was done, the functional assessment was performed?

7. Stroke location is critical, it may affect EEG and stroke recovery. The analysis should include it.

8. The definition of EEG TAR (parietal lesion side and central lesion side) needs be described.

9. The role of ICI and ICF need be described.

10. The sentence between lines 133 and 136, reference is necessary.

Results

11. Table 3, the confounding factors for adjusting should include stroke severity, stroke type, stroke location. ,

12. Location of stroke should be reported and included in the analysis.

13. Table 1 Abbreviation CMP should be defined.

14. ICI and ICF data and analysis were not reported.

15. Table 1. The patient number in race is incorrect and more than one race appears twice, and the number is inconsistence.

16. The data of EEG, TAR and rMT should be presented. How many patients received EEG or rMT should be presented.

Discussion

17. This is a prospective study and enrolls 102 participants. Why only 39 to 55 participants have motor function assessment data? It is important to state how the data will be analyzed.

Round 2

Reviewer 1 Report

Comments and Suggestions for Authors

I thank the Authors for their extensive work, which greatly improved the quality of their paper. No further comments. 

Reviewer 2 Report

Comments and Suggestions for Authors

Most of my comments have been addressed. However, minor revision is necessary.

1. Line 115, the word “includedif” may be misspelled.

2. Exclusion criteria were reported from line 118 to line 120. The exclusion criteria were repeated from line 122 to line 125. The exclusion criteria do not need to be repeated.

Author Response

All comments were revised.